# Versatile and efficient genome editing with *Neisseria cinerea* Cas9

Zhiquan Liu[1,6], Siyu Chen[1,6], Wanhua Xie[2], Hao Yu[1], Liangxue Lai ●[1,3,4,5 ✉] & Zhanjun Li ●[1 ✉]

The CRISPR/Cas9 system is a versatile genome editing platform in biotechnology and therapeutics. However, the requirement of protospacer adjacent motifs (PAMs) limits the genome targeting scope. To expand this repertoire, we revisited and engineered a compact Cas9 orthologue derived from *Neisseria cinerea* (NcCas9) for efficient genome editing in mammal cells. We demonstrated that NcCas9 generates genome editing at target sites with N4GYAT (Y = T/C) PAM which cannot be recognized by existing Cas9s. By optimizing the NcCas9 architecture and its spacer length, editing efficacy of NcCas9 was further improved in human cells. In addition, the NcCas9-derived Base editors can efficiently generate base conversions. Six anti-CRISPR (Acr) proteins were identified as off-switches for NcCas9. Moreover, NcCas9 successfully generated efficient editing of mouse embryos by microinjection of NcCas9 mRNA and the corresponding sgRNA. Thus, the NcCas9 holds the potential to broaden the CRISPR/Cas9 toolsets for efficient gene modifications and therapeutic applications.

[1] Key Laboratory of Zoonosis Research, Ministry of Education, College of Animal Science, Jilin University, Changchun 130062, China. [2] The Precise Medicine Center, Shenyang Medical College, Shenyang 110000, China. [3] CAS Key Laboratory of Regenerative Biology, Guangdong Provincial Key Laboratory of Stem Cell and Regenerative Medicine, South China Institute for Stem Cell Biology and Regenerative Medicine, Guangzhou Institutes of Biomedicine and Health, Chinese Academy of Sciences, Guangzhou 510530, China. [4] Guangzhou Regenerative Medicine and Health Guang Dong Laboratory (GRMH-GDL), Guangzhou 510005, China. [5] Institute for Stem Cell and Regeneration, Chinese Academy of Sciences, Beijing 100101, China. [6] These authors contributed equally: Zhiquan Liu, Siyu Chen. ✉email: lai_liangxue@gibh.ac.cn; lizj_1998@jlu.edu.cn

CRISPR-Cas9 system and derived genomic manipulation tools have transformed biotechnology and medicine[1,2]. They are composed of the Cas9 nucleases and a single guide RNA (sgRNA). Cas9 binds to the ~20 nucleotide (nt) spacer sequence and scaffold of sgRNA, and typically cleaves target DNA about 3 bp upstream of the PAM[3,4]. The CRISPR-guided DNA base editors (BEs), comprising deaminases fused to the N-terminus of Cas9 nickase (nCas9), enable targeted C-to-T (CBE) or A-to-G (ABE) conversions in genomic DNA[5,6]. The BEs, which can precisely install targeted point mutations without requiring DSBs or donor DNA templates, have exhibited powerful genome manipulation capability in various organisms[7,8].

To date, a series of natural Cas9 orthologs and artificially-designed Cas9 variants with various PAM requirements have been developed, such as SaCas9 (NNGRRT)[9], Nme1Cas9 (NNNNGATT)[10], Nme2-Cas9 (NNNNCC)[11], St1Cas9 (NNRGAA)[12], SpaCas9 (NNGYRA)[13], Cje1Cas9 (NNNVRYAC)[14], Cje3Cas9 (NNNNCYA)[15], SauriCas9 (NNGG)[16], SchCas9 (NNGR)[17], FrCas9 (NNTA)[18], ScCas9 (NNG)[19], SpCas9-VQR (NGA)[20], SpCas9-NG (NG)[21], SpCas9-NRNH (NRNH)[22], SpG (NG)[23], SpRY (NR > NY)[23], etc. These Cas9 nucleases expand the PAM compatibility of Cas9 orthologs and improve the flexibility of genome editing applications[24]. However, the NGG PAM requirement of conventional SpCas9 and relevant BEs, together with the complicated PAM requirements of small Cas enzymes, restrict the genome targeting scope. Therefore, it is necessary to explore new Cas9 nucleases with distinctive PAMs to expand the nuclease repertoire.

In this study, we revisited the attributes of *Neisseria cinerea* Cas9 (NcCas9), a small Cas9 orthologue, which were discovered yet has not been fully examined and utilized for genome editing in human cells[9]. NcCas9 was reported to be inefficient and recognized a restrictive N4GTA PAM, limiting its application in genome editing[9,25].

Here, we indicated that the engineered NcCas9 system enables efficient genome editing and base editing at endogenous sites with a distinct N4GYAT PAM, offering an alternative tool for both basic research and clinical applications.

## Results

### PAM identification of NcCas9 by PAM-DOSE. 
The type II-C Cas9 orthologue NcCas9 is of small size (1082 aa) and is closely related to conventional *Neisseria meningitidis* Cas9 (NmeCas9)[11] (Fig. 1a). The NcCas9 is 94% identical to Nme1Cas9, and the divergences lie mainly in the C-terminal PAM-interacting domain (PID) (Fig. 1a and S1). The NcCas9 locus also contains an array composed of 36-bp direct repeats (DRs) interspaced by 30-bp spacers in the proximity of the Cas genes operon, similar to that of Nme1/2Cas9 locus (Fig. 1a). Crystal structures of Nme1Cas9 and Nme2Cas9 have indicated that key residues of H1024/N1029 or D1028/R1033 determine the N4GATT or N4CC PAM specificity, respectively[26]. Interestingly, we noticed that the NcCas9 harbors PAM-interacting residues N1024 and A1029, different from that of Nme1/2Cas9 orthologs by protein sequence alignment (Fig. S1). Therefore, we speculated that the NcCas9 may recognize distinct PAM sequences from Nme1/2Cas9. The PAM of NcCas9 was identified as NNNNGTA using in vitro cleavage of a degenerate 7-bp sequence in the previous report[9]. However, many II-C Cas9 orthologues recognize an 8-nt PAM sequence (such as Nme1Cas9 (N4GATT) and CjeCas9 (N3VRYAC)) that cannot be fully characterized by the 7-bp library[11,14]. In addition, many Cas9 orthologues identified by library cleavage in vitro or in bacteria often do not function well in mammalian cells[9,27].

Therefore, we fully identified the functional PAMs of NcCas9 in human cells using a widely-used positive screening system

designated as PAM-DOSE (PAM Definition by Observable Sequence Excision)[28,29]. Briefly, a library of pmTmG plasmids containing a randomized 8-bp sequence (5′-NNNNNNNN-3′) was generated. After the cleavage of Nme1Cas9, Nme2Cas9 or NcCas9, the tdTomato cassette was excised, and the CAG promoter drove the expression of the EGFP gene[28]. Cleaved products were polymerase chain reaction (PCR) amplified and subjected to deep sequencing to identify PAM sequences recognized by these Cas9 nucleases (Fig. 1b). As expected, the conventional Nme1Cas9 and Nme2Cas9 recognized N4GATT and N4CC as optimal PAMs, respectively (Fig. 1c). In addition, a relaxed N4RYWT (R = A/G, Y = C/T, W = A/T) was recognized by NcCas9, which is different from that of Nme1Cas9, consistent with the previous speculation based on the divergent PAM-interacting residues (Fig. 1c). Importantly, the NcCas9 mainly recognized an N4GTAT PAM and showed an obvious preference for T in the eighth base, demonstrating that the previously identified N4GTA PAM was inaccurate (Fig. 1c).

### Detailed determination of efficient PAM sequences recognized by NcCas9. 
To further verify the PAM specificity of NcCas9, we performed a quantitative comparison by co-transfecting plasmids encoding Cas9, corresponding sgRNA, and pmTmG reporter plasmids containing variable PAM sequences based on an optimal N4GATT (Nme1Cas9), N4CC (Nme2Cas9), and N4GTAT (NcCas9) PAM (Fig. 2a). The editing efficiency of different PAMs was quantified by calculating the ratio of EGFP-positive cells to tdTomato-positive cells by flow cytometry analysis (Fig. 2a). As a result, Nme1Cas9 efficiently cleaved optimal N4GATT PAM and suboptimal N4GTTT PAM; Nme2Cas9 only potently recognized N4CC PAM (Fig. 2b, c). In contrast, reporter assays showed NcCas9 efficiently cleaved target sites containing optimal N4GTAT PAM and suboptimal N4GCAT PAM (Fig. 2b, c). In addition, the NcCas9 can also accept A or T at the fifth (N4ATAT) or seventh (N4GTTT) base of the PAM, respectively (Fig. 2b, c). At the eighth base, a strong preference for a T of NcCas9 was observed, and all other three bases significantly reduced editing efficiency (Fig. 2b, c). Importantly, despite the high sequence conservation, Nme1/2Cas9 could not cleave at primary NcCas9 PAMs (N4GYAT, where Y is T or C), highlighting distinct PAM specificity of the NcCas9 (Fig. 2b). These quantitative comparison results were consistent with the previous observations of PAM identification from PAM-DOSE. Taken together, these data demonstrated that NcCas9 can efficiently recognize distinct N4GYAT and suboptimal N4ATAT/N4GTTT PAMs, enriching Cas9 nucleases toolsets.

### Efficient genome editing in human cells by optimized NcCas9 system. 
In addition to previously-identified incomplete PAM sequences, the relatively low activity is what hinders NcCas9 from being widely used in the past[9,25]. Therefore, we attempted to improve its activity by optimizing the NcCas9 architecture and its sgRNA. First, the human codon-optimized ORF of NcCas9 was refined using codon usages from GenScript and flanked with bipartite nuclear localization signal (bpNLS), which have been proved to substantially improve the expression levels of BEs[30] (Fig. S2a). As a result, engineered NcCas9 showed a significant increase in editing efficiency compared to the previously-reported NcCas9 system in both pmTmG reporter assay and endogenous sites (Fig. S2b and S2c). In addition, we identified the sgRNA scaffold sequence used by NcCas9 and attempted to explore the optimal spacer length requirement (Fig. S3a and S3b). A series of sgRNAs targeting site5 with variable spacer lengths (20–27 nt) were tested. An additional guanine was added to the 5′-terminal sequence of each sgRNA to ensure

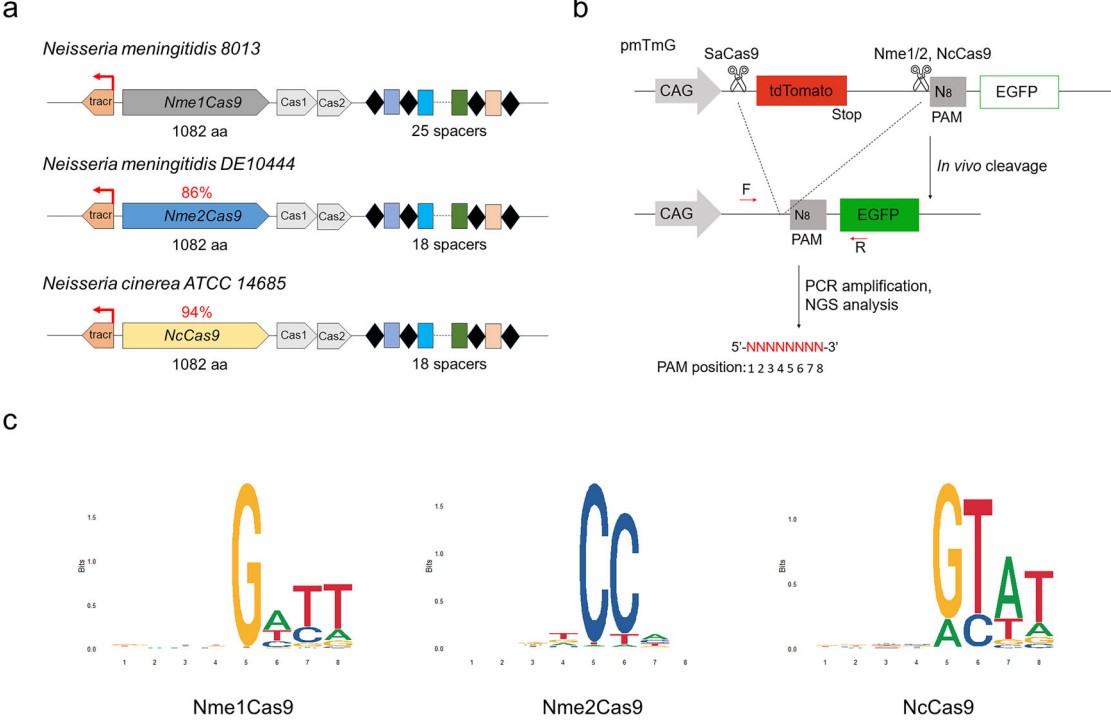

**Fig. 1 PAM identification by PAM-DOSE. a** Schematic showing the CRISPR/Cas loci of the strains encoding the Nme1Cas9, Nme2Cas9, and NcCas9 orthologs. Red arrows denote tracrRNA transcription initiation sites. Percent identities of Nme2Cas9 and NcCas9 with Nme1Cas9 are 86% and 94%, respectively. **b** Schematic showing the PAM-DOSE assay for characterization of PAM sequences recognized by Cas9s. An 8 N library of PAM sequences were introduced into the constructs. In the presence of a functional PAM, cleavage-mediated tdTomato cassette excision was performed, which led to the expression of EGFP. After Cas9 cleavage, cleaved products were PCR amplified and subjected to deep sequencing to identify PAM sequences. **c** The sequence logo showing the PAM specificity of Nme1Cas9, Nme2Cas9, and NcCas9 was obtained using deep-sequencing data.

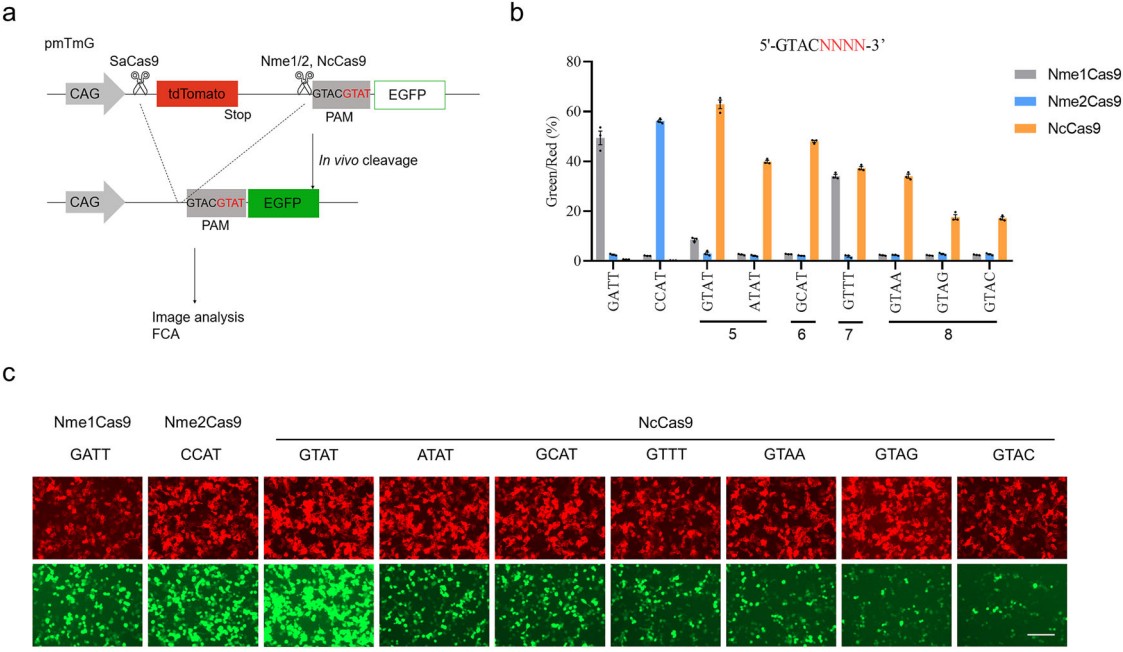

**Fig. 2 Detailed PAM sequence analysis of NcCas9. a** Schematic showing the pmTmG reporter assay for comparing different PAMs of Nme1Cas9, Nme2Cas9, and NcCas9 orthologs. Variable PAM sequences were introduced into the constructs. In the presence of a functional PAM, cleavage-mediated tdTomato cassette excision was performed, which led to the expression of EGFP, thereby allowing live visualization under a fluorescence microscope or detection via flow cytometry analysis (FCA). **b** Comparison of Nme1Cas9, Nme2Cas9, and NcCas9 cleavage efficiency of different PAMs by FCA. The different PAMs were designed based on the optimal GTACGTAT PAM sequence. Red or green represents the tdTomato or EGFP signal, respectively. **c** Representative fluorescence microscopy images of PAM analysis. Scale bar, 200 μm. Error bars indicate the s.e.m. ($n = 3$).

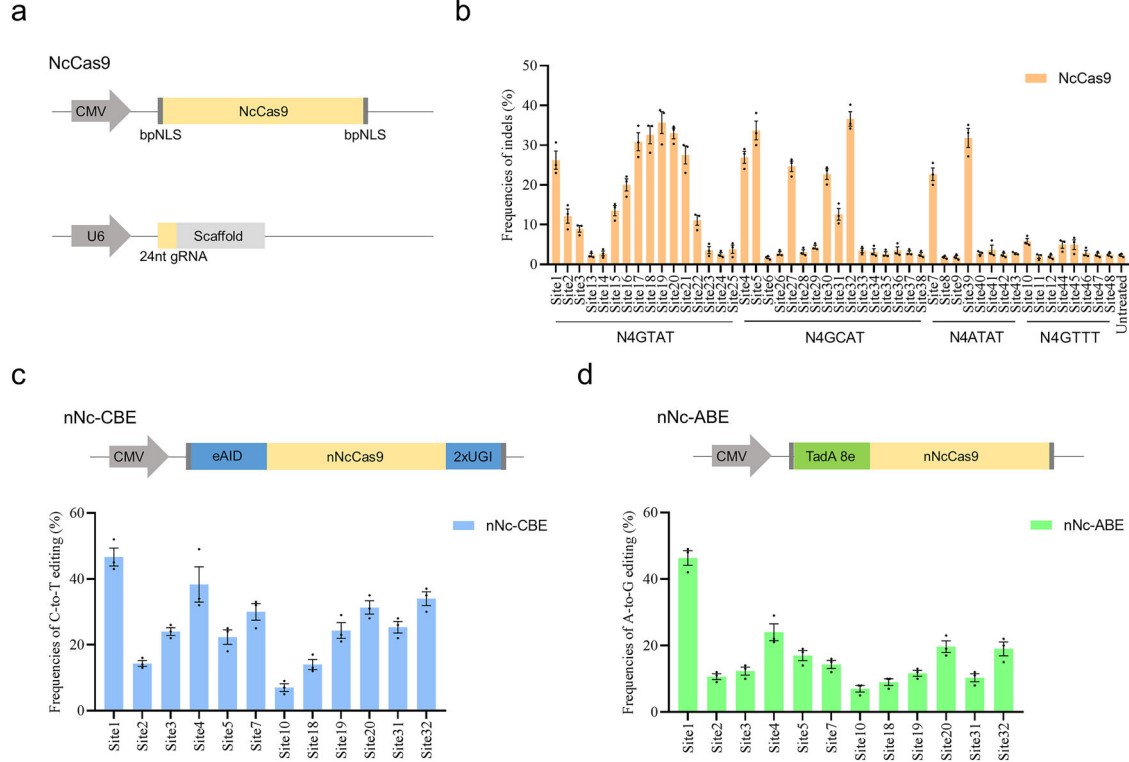

**Fig. 3 Genome editing of NcCas9 and its base editors in human cells. a** Schematic of optimized NcCas9 and its sgRNA architecture. **b** Genome editing of NcCas9 with N4GYAT, N4ATAT and N4GTTT PAMs in 48 endogenous loci. **c** Top: Schematic of the nNc-CBE. Bottom: Efficient C-to-T conversions in 12 endogenous loci by four nNc-CBE. **d** Top: Schematic of the nNc-ABE, TadA 8e was used. Bottom: Efficient A-to-G conversions in 12 endogenous loci by nNc-ABE. The highest C-to-T/A-to-G editing frequency within the editing window represents the base editing frequency at each site. Error bars indicate s.e.m. ($n = 3$).

proper transcription. We proved that 24 nt (G + 23) is the ideal spacer length, and activity decreased upon other lengths, indicating that the 20-nt spacer length of NcCas9 as utilized previously affected its editing activity[9,25] (Fig. S3c).

To thoroughly attest to the efficacy of NcCas9 for genome editing, we co-transfected the optimized NcCas9 system and corresponding sgRNAs with optimal 24-nt spacer length into HEK293T cells (Fig. 3a). A panel of 48 endogenous sites with functional PAMs including N4GTAT, N4GCAT, N4ATAT, and N4GTTT were tested (Fig. 3b and Table S2). As a result, obvious indels were observed at 25 out of 48 endogenous sites with varied editing efficiencies from 4.9% to 36.6% (Fig. 3b). Notably, NcCas9 generated efficient indels at target sites with N4GYAT (Y = T/C) PAMs, while showed lower efficiency at target sites with N4ATAT and N4GTTT PAMs (Fig. 3b). These endogenous gene editing results were consistent with the results of pmTmG reporter assays. In addition, we used the classical GUIDE-seq method to assess the on-target and off-target frequencies of NcCas9 targeting two different loci, Site1 and Site5[31,32]. As a result, no off-target cleavage was detected in the two sites (Fig. S4). Collectively, these results validated that the optimized NcCas9 system is a potent genome editing platform in human cells.

**Efficient base editing in human cells by NcCas9-derived BEs.** In addition to Cas9-mediated gene knockout, base editing is a revolutionary technology that can expediently induce targeted base conversions in desired sites without donor templates[7]. However, BEs are highly dependent on a proper PAM adjacent to the target base, limiting the targeting range[7,33]. Although many Cas9 variants have been used for base editing, more Cas9 nucleases with distinct PAMs are needed to enrich the BE tools. To test whether NcCas9

could be employed for base editing, we replaced the SpCas9 nickase in our previously-designed eAID-CBE system with NcCas9 nickase (D16A) and to create nNc-CBE (Fig. 3c). The eAID-CBE system has exhibited high editing activity and excellent Cas9 compatibility in the previous report[13,34,35]. Notably, the nNc-CBE showed efficient C-to-T editing in all 12 sites with average frequencies from 7.0% to 46.7% (Figs. 3c and S5). Encouraged by the results of CBE, we further generated nNc-ABE system with TadA 8e, the current optimal version of ABE[13,36] (Fig. 3d). As a result, the nNc-ABE also mediated the efficient conversion of A-to-G in genomic DNA with average frequencies from 7.0% to 46.3% (Figs. 3d and S6). Taken together, these data indicated that NcCas9 can be used to expand the targeting range of BEs.

**Identification of efficient Acr inhibitors for NcCas9.** Bacterial CRISPR–Cas systems utilize sequence-specific RNA-guided nucleases to defend against bacteriophage infection[37]. In response to the bacterial war on phage infection, numerous phages evolved anti-CRISPR (Acr) proteins to block the function of CRISPR-Cas systems[37,38]. As a natural brake for CRISPR/Cas nucleases, Acr has been widely used to manipulate genome editing in mammalian cells and organisms[38]. To test Acr inhibition of NcCas9, we performed NcCas9 and corresponding sgRNA transfections (targeting site5) in HEK293T cells in the presence or absence of 14 representatives Acr proteins (AcrIIA1-A6, A11, A13, A16, and AcrIIC1-C5). Consequently, AcrIIC3-C5 showed strong inhibition of NcCas9 genome editing, followed by AcrIIC1, C2, and A5, while others showed no obvious effect (Fig. 4a, b). Similar inhibitory effects of the six Acrs were observed in both nNc-CBE and nNc-ABE systems, indicating that Acrs can also be used to regulate the activity of nNc-BEs (Fig. 4c, d). Notably, the most robust

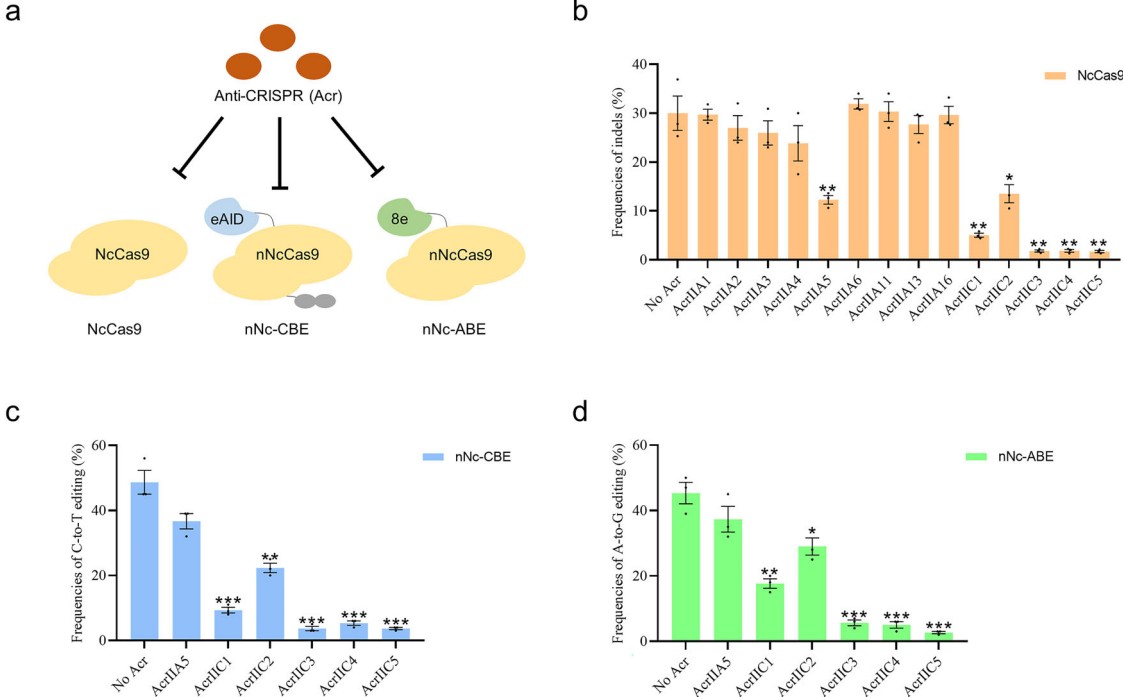

**Fig. 4 NcCas9 is inhibited by Acr proteins in human cells. a** Schematic of Acr proteins is used to inhibit the activities of NcCas9 and its base editors. **b** Genome editing of NcCas9 in the presence or absence of the 14 previously described Acr families. Plasmids expressing NcCas9, sgRNA targeting site5, and each Acr (1:1:1) were co-transfected into HEK293T cells. **c** Base editing of nNc-CBE in the presence or absence of six Acrs at the site1. **d** Base editing of nNc-ABE in the presence or absence of six Acrs at site1. Error bars indicate the s.e.m. ($n = 3$).

| Table 1 Generation of targeted editing in mouse embryos by NcCas9. | | | | | |
|---|---|---|---|---|---|
| **PAM** | **Target site** | **No. of zygotes** | **No. of blastocysts** | **No. of editing (%)** | **Frequencies of editing in single embryo (%)** |
| N4GTAT | Tyr-1 | 20 | 8 | 2 (25) | (26.1–30.7) |
| | Pcsk9-1 | 20 | 10 | 8 (80) | (21.2–98.6) |
| N4GCAT | Tyr-2 | 20 | 8 | 0 (0) | (0) |
| | Pcsk9-2 | 20 | 9 | 0 (0) | (0) |

inhibition was presented by AcrIIC3-C5 in both NcCas9 and its BEs, suggesting that they have a high potential fight against NcCas9 (Fig. 4b–d). These results suggested that the six Acr proteins can be used as off switches for NcCas9 applications.

**Genome editing in mouse embryos by NcCas9.** To evaluate the feasibility of NcCas9 genome editing in the mouse embryo, we selected four target sites with N4GYAT PAMs from *Tyr* and *Pcsk9* gene. Genome editing was performed in mouse zygotes by microinjection of NcCas9-encoding mRNA and the appropriate sgRNA, as previously reported[39,40]. After injection, mouse zygotes were cultured in vitro to blastocysts and then genotyped. Notably, two target sites with N4GTAT PAM showed efficient editing efficiency of 25% (Tyr-1) and 80% (Pcsk9-1) (Table 1). In addition, the editing frequencies in single embryo ranged from 26.1% to 30.7% in Tyr-1 and from 21.2% to 98.6% in Pcsk9-1 (Table 1). However, no obvious editing events were observed in two target sites with a suboptimal N4GCAT PAM (Table 1). These results demonstrated efficient genome editing in mouse embryos by NcCas9, underscoring the potential of the NcCas9 system in generating disease models in animals.

## Discussion

In this study, we identified a Cas9 orthologue derived from *N. cinerea* that can be utilized for efficient genome editing.

Importantly, we re-identified that NcCas9 recognizes a relaxed N4GYAT PAM, demonstrating that the previously identified N4GTA PAM was inaccurate. The editing efficiency of the NcCas9 system was improved through rational optimization of NcCas9 architecture and spacer length. We further validated that NcCas9 and NcCas9-derived BEs can potently perform gene editing in both human cells and mouse embryos. Efficient Acr inhibitors of NcCas9 were also developed for precise regulation and applications. Our study comprehensively characterized NcCas9 as a versatile and robust genome editing tool.

In addition, the compact size of NcCas9 (1082 aa) is compatible with the all-in-one-AAV vector for in vivo delivery. Although several small Cas9 nucleases have been used for genome editing via single-AAV delivery, the range of targetable sequences remains limited because of the PAM requirements. We demonstrated that NcCas9 recognized a novel PAM of N4GYAT, which is different from those of currently used small Cas9s, including SaCas9 (NNGRRT)[9], Nme1Cas9 (NNNNGATT)[10], Nme2Cas9 (NNNNCC)[11], CjeCas9 (NNNVRYAC)[14], St1Cas9 (NNRGAA)[12] and SpaCas9 (NNGYRA)[13]. Therefore, NcCas9 holds the potential to further expand the PAM compatibility of small Cas9 orthologues and improve the flexibility of therapeutic applications in the future.

We have validated that NcCas9-mediated CBE and ABE can induce robust C-to-T and A-to-G base editing. Recently, some new precise editing tools were generated, including glycosylase

BEs that can generate C-to-G transversions[41–43], dual BEs that can simultaneously catalyze both cytosine and adenine base conversions[44–47], and prime editors that can directly write new genetic information into a specified site[48]. The NcCas9 could be used as a new scaffold to expand the versatility of these precise genome editing systems in the future.

In summary, we developed a Cas9 ortholog, NcCas9, with a distinct N4GYAT PAM (N4GTAT > N4GCAT), that can induce efficient genome editing frequency of up to 36.6% in human cells. In addition, the NcCas9-derived CBE or ABE systems can efficiently generate targeted C-to-T or A-to-G base conversions of up to 46.7% or 46.3%, respectively. We also presented that AcrIIC3–C5 are robust inhibitors of both NcCas9 and its BEs, having the potential to be off-switches for NcCas9 applications. Moreover, the NcCas9 also successfully generated efficient editing in mouse embryos. Thus, the NcCas9 system is a promising tool for both basic research and therapeutic applications.

## Methods

**Animals**. C57/BL6 mice were obtained from the Laboratory Animal Center of Jilin University (Changchun, China). Superovulated C57BL/6J mice (6–8 week old females) were mated with C57BL/6J males. All mice and experimental protocols used in this project have been approved by the Institutional Animal Care and Use Committee of Jilin University (SY202111300).

**Plasmid construction**. The previous NcCas9, Nme1Cas9, and Nme2Cas9 was obtained from Addgene (#68331, #115694 and #119923). The pmTmG reporter plasmid was a kind gift from Feng Gu[28]. The human codon-optimized NcCas9 and its sgRNA scaffold were synthesized by Genscript Biotech (China). Plasmid site-directed mutagenesis was performed using the Fast Site-Directed Mutagenesis Kit (Tiangen, China). The site-directed mutation primers are listed in Table S1. The sequences of plasmids are listed in Supplementary Note 1.

**Cell culture and transfection**. HEK293T cell lines (obtained from ATCC and preserved by our laboratory) were cultured in Dulbecco's modified Eagle's medium (DMEM) supplemented with 10% fetal bovine serum (HyClone, China) and incubated at 37 °C in an atmosphere of 5% $CO_2$. The cells were seeded in 24-well plates and transfected using Hieff Trans$^{TM}$ Liposomal Transfection Reagent (Yeasen, China) according to the manufacturer's instructions. Puromycin (Meilunbio, China) was added at the final concentration of 3 μg/mL to enrich the positively transfected cells at 24 h after transfection. After 72 h, the cells were collected and used for genotyping by TIDE[49] or EditR[50]. All target sites and primers used for genotyping are listed in Tables S2 and S3.

**PAM characterization using PAM-DOSE**. The PAM-DOSE assay was carried out as previously reported[28]. Briefly, HEK293T cells were co-transfected with Nme1-Cas9/Nme2Cas9/NcCas9, corresponding sgRNA, SaCas9, and its sgRNA, and pmTmG library plasmids (1:1:1:1:1) using Hieff Trans$^{TM}$ Liposomal Transfection Reagent (Yeasen, China). PCR is used to amplify the sequences flanking the Cas9 target locus, and products were subjected to deep sequencing (Sangon Biotech, China). PAM regions without indel within three bases near PAM were extracted. PAMs were counted and used to generate sequence logos.

**pmTmG reporter assay**. HEK293T cells were co-transfected with Nme1Cas9/Nme2Cas9/NcCas9, corresponding sgRNA and pmTmG reporter plasmids (1:1:1) using Hieff Trans$^{TM}$ Liposomal Transfection Reagent. After 72 h of incubation, the fluorescent images were imaged with a microscope (ts100; Nikon, Tokyo, Japan). Cells were harvested for editing quantification by flow cytometry. Quantification was based on the relative fluorescent frequencies of green/red.

**GUIDE-seq analysis**. GUIDE-seq was constructed to assess the on-target and off-target frequencies of NcCas9 targeting Site1 and Site5 as previous studies reported[18,31,32]. GUIDE-seq and data analysis were completed by GeneRulor Company Bio-X Lab (Guangzhou, Guangdong, China). Briefly, 5 μl dsODN (100 μM) and 10 μg NcCas9-sgRNA plasmids were co-transfected into HEK293T cell line. After 72 h, cells were harvested for DNA extraction, followed by dsODN-PCR verification of effective cleavage. Then, GUIDE-seq libraries were constructed. The DNA went through shearing, adding Y adapters and two rounds of PCR, and was finally sequenced using MGISEQ-2000RS sequencer (PE150, paired-end) with customized settings for 16 bp UMI. Data were demultiplexed using in-house python scripts and then analyzed using guideseq v1.1[51]. No SpCas9 positive controls have been included to compare off-target reads in this study.

**mRNA and gRNA preparation**. The NcCas9, nNc-CBE, and nNc-ABE plasmids were linearized with NotI and transcribed in vitro using the HiScribe™ T7 ARCA mRNA kit (NEB). The mRNA was purified using the RNeasy Mini Kit (Qiagen) according to the manufacturer's protocol. The sgRNA oligos were annealed into the pUC57-Nc sgRNA expression vector containing a T7 promoter. The sgRNAs were then amplified and transcribed in vitro using the MAXIscript T7 kit (Ambion) and purified using the miRNeasy Mini Kit (Qiagen) according to the manufacturer's protocol.

**Microinjection of mouse zygotes and genotyping**. Briefly, a mixture of mRNA (50 ng/μl) and sgRNA (30 ng/μl) was co-injected into the cytoplasm of pronuclear-stage zygotes. Each group was injected with an average of approximately 20 zygotes to test the base editing efficiency. The injected zygotes were transferred to KSOM medium for culture at 37 °C, 5% $CO_2$, and 100% humidity. Then, the injected single zygote was collected at the blastocyst stage. Genomic DNA was extracted in embryo lysis buffer (1% NP40) at 56 °C for 60 minutes and then at 95 °C for 10 minutes in a Bio-Rad PCR Amplifier. Then, the extracted products were amplified by PCR (95 °C, 5 min for predegeneration, 42 cycles of (95 °C, 30 s, 58 °C, 30 s, 72 °C, 30 s), 72 °C, 5 min for extension) and determined by Sanger sequencing. The genomic DNA of newborn mice was extracted from ear clips and analyzed by PCR genotyping. All target sites and primers used for genotyping are listed in Tables S2 and S3.

**Statistics and reproducibility**. All data are expressed as mean ± s.e.m. of at least three individual determinations for all experiments. Data were analyzed by student's $t$-test via GraphPad prism software 8.0.1. The probability value that was smaller than 0.05 ($p < 0.05$) was statistically significant. *$p < 0.05$, **$p < 0.01$, ***$p < 0.001$.

**Reporting summary**. Further information on research design is available in the Nature Portfolio Reporting Summary linked to this article.

## Data availability

All data generated or analyzed during this study are included in this published article and its supplementary files. Deep sequencing data of the PAM-DOSE assay have been deposited at Sequence Read Archive under accession code SRP404972. Raw sequencing data of GUIDE-seq have been deposited at the Sequence Read Archive under accession code SRP398785. The constructed plasmids are available from addgene (accession IDs: 194093 to 194095). The source data are listed in Supplementary Data 1.

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

## Acknowledgements

This study was financially supported by the National Key Research and Development Program (Grant no. 2018YFB1404205) of China Stem Cell and Translational Research (Grant no. 2017YFA0105101) and the National Natural Science Foundation of China (Grant nos. 32170543 and 31970574).

## Author contributions

Z.Q.L., L.X.L., and Z.J.L. conceived and designed the experiments. Z.Q.L., S.Y.C., W.H.X., and H.Y. performed the experiments and analyzed the data. Z.Q.L. and S.Y.C. wrote the paper. All authors read and approved the final paper.

## Competing interests

The authors declare no competing interests.
