## [Peer Review File · Communications Biology]

Reviewers' comments:

Reviewer #1 (Remarks to the Author):

The manuscript by Chen et al demonstrated that *Neisseria cinerea* Cas9 (NcCas9) could be optimized for genome editing in cells and mouse embryos. While some previous studies showed that NcCas9 had limited genome editing efficiency, Chen et al. optimized NcCas9 by changing the codon usage and spacer length. They observed that the optimized NcCas9 recognized the PAM sequence of N4GTAT rather than N4GTA PAM that was previously reported. They showed that the optimized NcCas9 could be utilized for genome editing and base editing in human cells. The genome editing efficiency of NcCas9 could be significantly reduced by some of the known anti-CRISPR proteins such as AcrIIC3-C5. NcCas9 could also be utilized in mouse embryo genome editing by microinjection of NcCas9 encoding mRNA and guide RNAs. The study seems to be of potential interest to researchers interested in application of novel CRISPR systems to genome editing. It may further strengthen the manuscript if the authors could provide some analyses and/or discussion on the potential off-target effect of NcCas9 in genome editing applications.

Reviewer #2 (Remarks to the Author):

In this manuscript, Tang, et al. describe a novel Cas9 variant from *Neisseria cinerea* Cas9, and characterize its PAM preference, editing efficiency for both nuclease and base editing, and further identify anti-CRISPR proteins that can block its activity. Though the PAM of (N)4GYAT is not a very broad specificity, it has the potential to edit unique sequences in a PAM-specific fashion, thus contributing to its novelty. I also commend the authors on the mouse zygote assay, which provides a use-case for NcCas9 to generate appropriate disease models. Overall, I appreciate the concise and simple nature of the paper, but do believe multiple additional assays must be conducted to improve the paper. I highlight these suggestions below:

Main Revisions

1. This is the first use-case of the PAM-DOSE assay outside of the main paper (from what I gather). The current state-of-the-art in the PAM engineering field is the PAMDA assay. I request the authors conduct PAMDA on NcCas9 to see if it corroborates the PAM-DOSE assay results: <https://www.nature.com/articles/s41596-020-00465-2>
2. The authors only test 12 sites for indel efficiency and 6 sites for base editing. Since the Cas9 is a brand new variant, I ask that the authors test 48 different target sequences in at least 8-12 different genomic loci for indel formation, and 12 targets for each form of base editing. This will give readers a better understanding of NcCas9 activity for downstream usage. The current results, however, are promising, and so I look forward to seeing editing on more sites.
3. The authors present no data on off-targeting, which is a requirement for any new Cas9 profile. I request performance of the GUIDE-Seq assay (<https://www.nature.com/articles/nbt.3117>) on two different sites to verify the off-target profile of NcCas9.

Minor Comments

1. I request that the authors number their references in the order presented in the text, so it's easier to read.
2. I would also ask the authors to do a deeper dive into the PAM Cas9 literature for the introduction and discussion. Please use this paper (<https://www.nature.com/articles/s41467-020-20633-y>) as a guide to cite more relevant literature on Cas9 orthologs and variants.

Reviewer #3 (Remarks to the Author):

Comments to the Author

The manuscript by Chen et al. is a research article entitled "Versatile and efficient genome editing with *Neisseria cinerea* Cas9". This is a very excellent research article as timely written with novelty. The authors explored and identified the small NcCas9 for in vivo genome editing as an alternative tool for basic research and clinical applications. Here are some comments:

1. The introduction must be improved. Present the state of the art in this section. It should include more recent trends, and currently known research findings.
2. The caption of figure 1 must be elaborated.
3. Conclusions must be elaborated and represented in more detail.

Based on the following comments, the manuscript is suitable for publication in *Communications biology*.

Dear reviewers:

Thank you very much for your comments concerning our manuscript entitled “Versatile and efficient genome editing with *Neisseria cinerea* Cas9 (COMMSBIO-22-0820)”. Those comments are all valuable and very helpful for revising and improving our paper, as well as the important guiding significance to our researches. We have revised the manuscript accordingly (with blue fonts in the text) and a detailed response to the reviewers’ comments has been provided below.

Responses to Reviewer’s Comments:

To Reviewer #1:

1. * It may further strengthen the manuscript if the authors could provide some analyses and/or discussion on the potential off-target effect of NcCas9 in genome editing applications.

Response:

Thank you for your kind suggestion. The off-target effects of NcCas9 have been evaluated at two different sites by GUIDE-seq analysis. The results have been added in Fig. S4 and discussed in line 164-167 of the revised manuscript accordingly.

To Reviewer #2:

1. * This is the first use-case of the PAM-DOSE assay outside of the main paper (from what I gather). The current state-of-the-art in the PAM engineering field is the PAMDA assay. I request the authors conduct PAMDA on NcCas9 to see if it corroborates the PAM-DOSE assay results: <https://www.nature.com/articles/s41596-020-00465-2>

Response:

Thank you for your kind suggestion. Actually, the PAM-DOSE assay, an effective method for PAM identification, has been widely used to identify PAM sequences of Cas orthologs, including SpCas9, Cas12a, SpRY Cas9, SpaCas9 and Cje3Cas9^{1, 2, 3, 4}.

In addition to PAM-DOSE assay, the efficient PAM sequences of NcCas9 were further determined by flow cytometry analysis and endogenous genomic editing (Fig. 2 and 3b).

2. * The authors only test 12 sites for indel efficiency and 6 sites for base editing. Since the Cas9 is a brand new variant, I ask that the authors test 48 different target sequences in at least 8-12 different genomic loci for indel formation, and 12 targets for each form of base editing. This will give readers a better understanding of NcCas9 activity for downstream usage. The current results, however, are promising, and so I look forward to seeing editing on more sites.

Response:

Thank you for your kind suggestion. A total of 48 targets in 8 different genomic loci for indel

formation, and 12 targets for CBE/ABE have been conducted. The results have been added in Fig. 3b-3d, S5, S6 and Table S2 of the revised manuscript accordingly.

3. * The authors present no data on off-targeting, which is a requirement for any new Cas9 profile. I request performance of the GUIDE-Seq assay (<https://www.nature.com/articles/nbt.3117>) on two different sites to verify the off-target profile of NcCas9.

Response:

Thank you for your kind suggestion. The GUIDE-seq assay on two different sites of NcCas9 have been performed, and the results have been added in Fig. S4 and discussed in line 164-167 of the revised manuscript accordingly.

4. * I request that the authors number their references in the order presented in the text, so it's easier to read.

Response:

Thank you for your kind suggestion. We have numbered the references in order of appearance in revised manuscript accordingly.

5. * I would also ask the authors to do a deeper dive into the PAM Cas9 literature for the introduction and discussion. Please use this paper (<https://www.nature.com/articles/s41467-020-20633-y>) as a guide to cite more relevant literature on Cas9 orthologs and variants.

Response:

Thank you for your kind suggestion. The relevant literatures on Cas9 orthologs and variants have been cited in line 76-86 of the revised manuscript accordingly.

To Reviewer #3:

1. * The introduction must be improved. Present the state of the art in this section. It should include more recent trends, and currently known research findings.

Response:

Thank you for your kind suggestion. The recent trends and research findings have been added in line 76-86 of the introduction accordingly.

2. * The caption of figure 1 must be elaborated.

Response:

Thank you for your kind suggestion. The caption of Figure 1 has been elaborated in the legend of Figure 1 accordingly.

3. * Conclusions must be elaborated and represented in more detail.

Response:

Thank you for your kind suggestion. The Conclusions have been elaborated in line 234-239 of the

revised manuscript accordingly.

References

1. Tang L, *et al.* Efficient cleavage resolves PAM preferences of CRISPR-Cas in human cells. *Cell regeneration (London, England)* **8**, 44-50 (2019).
2. Ye J, *et al.* Can SpRY recognize any PAM in human cells? *Journal of Zhejiang University Science B* **23**, 382-391 (2022).
3. Liu Z, *et al.* Versatile and efficient in vivo genome editing with compact *Streptococcus pasteurianus* Cas9. *Molecular therapy : the journal of the American Society of Gene Therapy* **30**, 256-267 (2022).
4. Chen S, Liu Z, Xie W, Yu H, Lai L, Li Z. Compact Cje3Cas9 for Efficient In Vivo Genome Editing and Adenine Base Editing. *Crispr j* **5**, 472-486 (2022).

Reviewers' comments:

Reviewer #1 (Remarks to the Author):

Previous comments for author and rebuttal

To Reviewer #1:

1. * It may further strengthen the manuscript if the authors could provide some analyses and/or discussion on the potential off-target effect of NcCas9 in genome editing applications.

Response:

Thank you for your kind suggestion. The off-target effects of NcCas9 have been evaluated at two different sites by GUIDE-seq analysis. The results have been added in Fig. S4 and discussed in line 164-167 of the revised manuscript accordingly.

Comments for the authors in the revised manuscript.

The authors addressed the reviewer's previous comments by investigating the off-target effects of *Neisseria cinerea* Cas9 (NcCas9) by conducting GUIDE-seq assay. To this end, the authors analyzed the on-target indel rates of 48 sgRNAs and selected two (target 1, and target 5) that showed relatively high indel rates. Next the authors conducted GUIDE-seq for the two sites and only the on-target reads were detected in both assays (34892 reads for target 1, and 12951 reads for target 5). The results seem to be consistent with the hypothesis that the NcCas9 has relatively low off-target activity.

Minor comments.

line 276: It might be helpful if the authors could provide more details of how the GUIDE-seq assays were conducted in the study, such as the experimental conditions and bioinformatic analyses.

Reviewer #2 (Remarks to the Author):

The authors have responded to my concerns. As a minor point, I request the authors provide their PAM-DOSE plasmids on Addgene before publication.

Dear reviewers:

Thank you very much for your comments concerning our manuscript entitled “Versatile and efficient genome editing with *Neisseria cinerea* Cas9 (COMMSBIO-22-0820A)”. Those comments are all valuable and very helpful for revising and improving our paper, as well as the important guiding significance to our researches. We have revised the manuscript accordingly (with blue fonts in the text) and a detailed response to the reviewers’ comments has been provided below.

Responses to Reviewer’s Comments:

To Reviewer #1:

1. * The authors addressed the reviewer's previous comments by investigating the off-target effects of *Neisseria cinerea* Cas9 (NcCas9) by conducting GUIDE-seq assay. To this end, the authors analyzed the on-target indel rates of 48 sgRNAs and selected two (target 1, and target 5) that showed relatively high indel rates. Next the authors conducted GUIDE-seq for the two sites and only the on-target reads were detected in both assays (34892 reads for target 1, and 12951 reads for target 5). The results seem to be consistent with the hypothesis that the NcCas9 has relatively low off-target activity.

Minor comments.

line 276: It might be helpful if the authors could provide more details of how the GUIDE-seq assays were conducted in the study, such as the experimental conditions and bioinformatic analyses.

Response:

Thank you for your kind suggestion. The experimental conditions and bioinformatic analyses of the GUIDE-seq assays have been added in line 276-288 of the revised manuscript accordingly.

To Reviewer #2:

1. * The authors have responded to my concerns. As a minor point, I request the authors provide their PAM-DOSE plasmids on Addgene before publication.

Response:

Thank you for your kind suggestion. The PAM-DOSE method was previously developed by Feng Gu group¹. The PAM-DOSE plasmids used in this study were kind gifts from Feng Gu. It has been declared in line 249 of the revised manuscript.

References

1. Tang L, *et al.* Efficient cleavage resolves PAM preferences of CRISPR-Cas in human cells. *Cell regeneration (London, England)* **8**, 44-50 (2019).

REVIEWERS' COMMENTS:

Reviewer #1 (Remarks to the Author):

The authors seem to have address the reviewers' concerns. Some minor points that might help further clarify the manuscript is described below.

Line 266: The information of the reagent company of Hieff TransTM Liposomal Transfection Reagent could be added using the format of (company name, Country) as line 268 or line 274

Line 279: the company information could be formatted in parenthesis such as: (Guangzhou, Guangdong, China)

Line 279: It might be helpful to add of the concentration (in uM or nM) of the 5 ul dsODN used in the Guide-seq assay.

Line 283: It might be helpful to add the length (50 bp, 100bp, 150 bp or other) and the type (paired or single end) of the deep sequencing reads, such as 150 bp paired end.

Dear reviewers:

Thank you very much for your comments concerning our manuscript entitled “Versatile and efficient genome editing with *Neisseria cinerea* Cas9 (COMMSBIO-22-0820B)”. Those comments are all valuable and very helpful for revising and improving our paper, as well as the important guiding significance to our researches. We have revised the manuscript accordingly (with blue fonts in the text) and a detailed response to the reviewers’ comments has been provided below.

Responses to Reviewer’s Comments:

1. *

Line 266: The information of the reagent company of Hieff Trans™ Liposomal Transfection Reagent could be added using the format of (company name, Country) as line 268 or line 274

Response:

Thank you for your kind suggestion. The reagent company has been added in line 267 of the revised manuscript accordingly.

2. *

Line 279: the company information could be formatted in parenthesis such as: (Guangzhou, Guangdong, China)

Response:

Thank you for your kind suggestion. The company information has be formatted in line 279 of the revised manuscript accordingly.

3. *

Line 279: It might be helpful to add of the concentration (in uM or nM) of the 5 ul dsODN used in the Guide-seq assay.

Response:

Thank you for your kind suggestion. The concentration of the 5 ul dsODN (100uM) has been added in line 280 of the revised manuscript accordingly.

4. *

Line 283: It might be helpful to add the length (50 bp, 100bp, 150 bp or other) and the type (paired or single end) of the deep sequencing reads, such as 150 bp paired end.

Response:

Thank you for your kind suggestion. The length and type of the deep sequencing reads (PE150, paired end) have been added in line 284 of the revised manuscript accordingly.